# A Conserved Receptor-Binding Domain in the VP1u of Primate Erythroparvoviruses Determines the Marked Tropism for Erythroid Cells

**DOI:** 10.3390/v14020420

**Published:** 2022-02-17

**Authors:** Cornelia Bircher, Jan Bieri, Ruben Assaraf, Remo Leisi, Carlos Ros

**Affiliations:** 1Department of Chemistry, Biochemistry and Pharmaceutical Sciences, University of Bern, 3012 Bern, Switzerland; cornelia.bircher@uzh.ch (C.B.); jan.bieri@unibe.ch (J.B.); ruben.assaraf@unibe.ch (R.A.); remo.leisi@cslbehring.com (R.L.); 2CSL Behring AG, 3000 Bern, Switzerland

**Keywords:** parvovirus B19, B19V, VP1u, VP1uR, receptor, tropism, primate erythroparvovirus, simian erythroparvovirus, rhesus erythroparvovirus, pig-tailed erythroparvovirus

## Abstract

Parvovirus B19 (B19V) is a human pathogen with a marked tropism for erythroid progenitor cells (EPCs). The N-terminal of the VP1 unique region (VP1u) contains a receptor-binding domain (RBD), which mediates virus uptake through interaction with an as-yet-unknown receptor (VP1uR). Considering the central role of VP1uR in the virus tropism, we sought to investigate its expression profile in multiple cell types. To this end, we established a PP7 bacteriophage-VP1u bioconjugate, sharing the size and VP1u composition of native B19V capsids. The suitability of the PP7-VP1u construct as a specific and sensitive VP1uR expression marker was validated in competition assays with B19V and recombinant VP1u. VP1uR expression was exclusively detected in erythroid cells and cells reprogrammed towards the erythroid lineage. Sequence alignment and in silico protein structure prediction of the N-terminal of VP1u (N-VP1u) from B19V and other primate erythroparvoviruses (simian, rhesus, and pig-tailed) revealed a similar structure characterized by a fold of three or four α-helices. Functional studies with simian parvovirus confirmed the presence of a conserved RBD in the N-VP1u, mediating virus internalization into human erythroid cells. In summary, this study confirms the exclusive association of VP1uR expression with cells of the erythroid lineage. The presence of an analogous RBD in the VP1u from non-human primate erythroparvoviruses emphasizes their parallel evolutionary trait and zoonotic potential.

## 1. Introduction

Human parvovirus B19 (B19V), also known as primate erythroparvovirus 1, is a small, nonenveloped icosahedral virus classified within the genus *Erythroparvovirus* of the family *Parvoviridae* [1]. B19V infections are typically associated with the childhood rash disease *erythema infectiosum*, also known as fifth disease [2]. In adults, B19V infection causes an expanding range of syndromes, and factors influencing the severity of the infection are poorly understood. In individuals with underlying immune or hematologic disorders, B19V may cause severe cytopenias, myocarditis, vasculitis, glomerulonephritis, and encephalitis [3]. Although rare, B19V may also cause lethal infections [4]. The infection has been frequently associated with arthropathies in adults and represents a risk factor for maternal–fetal transmission, causing fetal anemia, non-immune fetal hydrops, and fetal death [5,6]. B19V is transmitted primarily through the respiratory route by aerosol droplets [7]. The virus can also be transmitted vertically to the developing fetus, via blood transfusion, contaminated plasma-derived therapeutic products, and organ transplantation [8,9,10,11]. Following the main entry through the respiratory route, the virus targets and productively exclusively infects erythroid precursor cells (EPCs) in bone marrow. The damage caused to the infected EPCs leads to the erythroid disorders during infection [12]. The internalization of B19V by antibody-dependent enhancement [13,14] may explain the presence of viral components in nonerythroid tissue [15], as well as the possible association of B19V infection with cardiovascular disease [16,17].

The nonenveloped B19V capsid is a compact T = 1 icosahedral particle composed of two structural proteins, VP1 and VP2, that share the same C-terminal sequence [18]. VP1 has an additional N-terminal extension, the so-called “VP1 unique region” (VP1u). The most N-terminal region of VP1u harbors a cluster of epitopes that are targeted by neutralizing antibodies [19,20,21], indicating the importance of this region in viral infection. VP1u is not accessible to antibodies in virions circulating in the blood [22]; however, this region becomes accessible upon interaction with the target cells, a process that is required for virus uptake [23,24]. In earlier studies, we revealed that this region specifically binds a receptor, herein named VP1uR, which is required for B19V uptake and productive infection [25]. VP1uR is expressed in erythroid cells and coincides with the homing of BFU-E cells to erythroblastic islands in bone marrow, as well as the subsequent differentiation to the immobilized CFU-E stage, proerythroblasts and early basophilic erythroblasts [26]. The receptor-binding domain (RBD) in the N-terminal region of VP1u spans amino acids 5-80. By introducing specific mutations, we identified the critical residues required for a functional RBD. Structural predictions and mutational studies reveal an RBD consisting of a spatial configuration of three consecutive α-helices. Although VP1u monomers were functional, their dimerization promoted receptor interaction and internalization [27].

The genus *Erythroparvovirus* comprises three additional non-human primate viruses, i.e., primate erythroparvovirus 2 (simian parvovirus (SPV)); primate erythroparvovirus 3 (rhesus macaque parvovirus (RhMPV)); and primate erythroparvovirus 4 (pig-tailed macaque parvovirus (PtMPV)). SPV was identified in cynomolgus monkeys with anemia [28,29]. PtMPV and RhMPV were identified in anemic pig-tailed macaques and rhesus monkeys, respectively [30]. The genomic organization of all primate erythroparvoviruses is very similar. The predicted non-structural protein 1 (NS1) and VP1 of SPV share 50% and 62% similarity with B19V, respectively. Similar to B19V, there is a putative splice acceptor site for the VP2 and 67% homology with B19 VP2. Equivalent protein sizes and homologies are estimated for RhMPV and PtMPV based on sequence information [31].

Epidemiologic studies suggest that SPV transmission occurs horizontally, probably through the respiratory tract, as seen for B19 infections in humans [32]. Experimental infection of cynomolgus macaques with SPV produced transient viremia, reticulocytopenia, and typical changes of erythroid cells in the bone marrow [33]. When SPV was discovered, the affected macaques were coinfected with immunosuppressive type D simian retrovirus (SRV). Only macaques dually infected with SPV and SRV became anemic [32], suggesting that most SPV infections are asymptomatic, which complicates the determination of their prevalence. Infection rates for both PtMPV and RhMPV are also undetermined [31]. The significant anemia induced by SPV, PtMPV, and RhMPV in immunosuppressed animals suggests a similar tropism for erythroid progenitor cells in bone marrow, as observed for B19V. In line with this hypothesis, SPV showed high tropism for erythroid progenitor cells, as identified by in vitro infection of cynomolgus bone marrow cells [31].

SPV VP2 immunoblotting of sera from monkey handlers appeared to indicate a correlation between exposure to SPV-positive macaques and the presence of SPV antibodies in human blood [34]. However, possible cross-reactivity with B19V antibodies has not been excluded. SPV can replicate, although moderately, in human bone marrow in vitro [34], and evidence of B19V replication in cynomolgus bone marrow cells was revealed by in situ hybridization and immunofluorescent microscopy [35]. These experimental observations suggest that SPV and probably RhMPV and PtMPV may be able to infect humans.

Considering the important function of VP1u-RBD in B19V tropism and cell entry, we sought to examine the expression profile of the VP1u cognate receptor in cells from different tissues and to explore the presence of an analog motif in non-human primate erythroparvoviruses. In previous studies, we used a recombinant VP1u and an empty MS2 bacteriophage-VP1u bioconjugate to detect VP1uR expression in various cell types by immunofluorescence [25,26,36]. In this study, we incorporated the VP1u region of B19V on the surface of *Pseudomonas aeruginosa* bacteriophage PP7 by click chemistry. The PP7-VP1u bioconjugate shares the size and VP1u composition of native B19V but lacks the entire VP2. This approach allowed for a more sensitive and quantitative determination of the expression profile of VP1uR in multiple cell types, as well as the possibility to study VP1uR-dependent virus uptake by RT-qPCR. This study confirms the strong association of VP1uR expression with cells of the erythroid lineage and identifies an equivalent RBD in the N-VP1u from non-human primate erythroparvoviruses.

## 2. Materials and Methods

### 2.1. Cells, Viruses, and Bacteria

UT7/Epo cells were cultured in MEM with 5% fetal calf serum (FCS) and 2 U/mL recombinant human erythropoietin (Epo). KU812Ep6 cells were cultured in RPMI 1640 with 10% FCS and 6 U/mL Epo. HepG2 and MRC-5 cells were cultured in DMEM with 10% FCS. K562 and KG1a cells were cultured in IMDM with 10% FCS. HeLa, HEK 293, NB324K, and A549 cells were cultured in DMEM with 5% FCS. REH cells were cultured in RPMI 1640 with 20% FCS. Erythroid progenitor cells (EPCs) were cultured in IMDM supplemented with 20% BIT 9500, 100 ng/mL SCF, 5 μg/mL IL-3, 1 µM hydrocortisone, and 3 U/mL rhEpo to induce differentiation toward the erythroid lineage. HiDEP and HuDEP cells were cultured in IMDM with 15% BIT 9500, 50 ng/mL SCF, 3 U/mL Epo, 1 µM dexamethasone, and 1 μg/mL doxycycline. HUVEC cells were cultured in vascular cell basal medium (ATCC PCS-100-030) supplemented with the endothelial cell growth kit-VEGF (ATCC PCS-100-041) containing 5 ng/mL rh VEGF, 5 ng/mL rh EGF, 5 ng/mL rh FGF basic, 15 ng/mL rh IGF-1, 10 mM L-glutamine, 0.75 U/mL heparin sulfate, 1 µg/mL hydrocortisone, 50 µg/mL ascorbic acid, and 2% FCS. Human dermal fibroblasts were cultured in fibroblast basal medium (ATCC PCS-201-030) supplemented with fibroblast growth kit-serum-free (ATCC PCS-201-040) containing 500 µg/mL human serum albumin (HSA), 0.6 mM linoleic acid, 0.6 µg/mL lecithin, 7.5 mM L-glutamine, 5 ng/mL rh FGF basic, 5 ng/mL rh EGF/TGF-1 supplements, 5 µg/mL rh insulin, 1 µg/mL hydrocortisone, and 50 µg/mL ascorbic acid. All culture media were supplemented with L-glutamine and 50 U/mL penicillin-streptomycin. B19V- and B19-infected human plasmas were obtained from CSL Behring AG, Charlotte, NC, USA. *Pseudomonas aeruginosa* and *Pseudomonas aeruginosa* bacteriophage PP7 were obtained from ATCC; 15692 and 15692-B4, respectively.

### 2.2. PP7 production

A volume of 100 μL of densely grown *P. aeruginosa* starter culture was infected with 0.1 μL of a PP7 stock in 18 mL soft agar (37 °C), composed of LB broth and LB agar Miller (7.5 g/l agar) in a 1:1 ratio. The inoculated soft agar was poured on a Petri dish (165 cm^2^) and incubated overnight at 37 °C. The soft agar was scraped off and transferred to a 50 mL falcon tube. After the addition of 10 mL PBS, the tube was vortexed and centrifuged at 3000× *g* for 25 min at RT. The supernatant was transferred to a new tube and centrifuged again at 3000× *g* for 10 min. The supernatant, containing the PP7, was filtered through a 0.22 μm filter. In the last step, PP7 was purified by ultracentrifugation through a 20% sucrose cushion (150,000× *g*, 4 h at 4 °C). Plaque-forming units and protein concentrations were quantified by plaque assay and SDS-PAGE, respectively.

Plaque assays of PP7 were carried out with the host bacteria *P. aeruginosa*. A volume of 10 μL of a densely grown bacteria starter culture was mixed with 2.5 mL soft agar and poured onto a prepared agar plate. Ten-fold dilutions of PP7 were prepared in LB broth and then evenly distributed on top of the solidified soft agar. The plates were incubated overnight at 37 °C. Plaques were counted the next day, and the results were expressed as plaque-forming units per milliliter (PFU/mL).

### 2.3. B19V and Simian VP1u Expression

B19V VP1u cloning and expression were carried out as previously described [26]. The DNA fragment-encoding simian VP1u was amplified from a pET-21(+) vector (Twist Bioscience, San Francisco, CA, USA) with restriction-site overhang primers (*Hin*dIII forward, 5′-CGA AAG CTT AGT GAG CCT GCT TCC AAA A-3′; *Kpn*I reverse, 5′-CGA GGT ACC ACA GTC TCT AAC CAC TTG T-3′). The amplicon was cloned into the pT7-FLAG-MAT Tag-2 expression vector (Sigma, St Louis, MO, USA), verified by sequencing, and transformed into *E. coli* XL10 Gold. Protein expression was carried out in *E. coli* BL21(DE3) cells induced with 1 mM isopropyl-β-D-thiogalactopyranoside (IPTG) at an OD_600_ of 0.6 for 4 h at 37 °C. The recombinant VP1u was purified twice with Ni-NTA magnetic agarose beads (Invitrogen, Carlsbad, CA, USA) under native conditions.

### 2.4. PP7-VP1u Bioconjugation

B19V and simian VP1u were bioconjugated to PP7 capsids following the same protocol. The bifunctional crosslinker trans-cyclooctene-PEG_4_-NHS ester (TCO-NHS; Jena Bioscience, Thuringia, Germany) was dissolved in DMSO and added to the PP7 capsids (2 mg/mL) to a final concentration of 50 μM. Crosslinking between the NHS ester and accessible lysines was carried out for 1 h at room temperature in PBS. Subsequently, lysine-reactive fluorescent dye NHS-Atto488 was dissolved in DMSO and added to the PP7-TCO capsids to obtain a final concentration of 100 μM. The fluorescent-labelling reaction was carried out for 1 h at room temperature in PBS (pH 7.4). The reaction was quenched with the addition of 50 mM Tris (pH 8). Non-coupled crosslinker and fluorescent dye were separated from the TCO capsids by centrifugation through 20% sucrose (150,000× *g*, 2 h, 4 °C). In a second step, VP1u was reduced with 1 mM TCEP. After reduction, TCEP was removed using a 0.5 mL Amicon 10 kDa filter device. The bifunctional crosslinker methyl-tetrazine-PEG_4_-maleimide (MeTz-Mal) was dissolved in DMSO and added to the previously reduced VP1u in a 20x molar excess. The crosslinking reaction was incubated for 2h at room temperature. Non-coupled crosslinker was separated from VP1u-MeTz with a 0.5 mL Amicon 10 kDa filter (washed 3x with PBS). The concentration of VP1u-MeTz and PP7-TCO was quantified with an SDS-PAGE, as well as Coomassie staining and densitometric analysis. The final click reaction between PP7-TCO and VP1u-MeTz was carried out by incubation of PP7-TCO with a 5x molar excess of VP1u-MeTz for 1h at room temperature. Non-coupled VP1u-MeTz was separated from PP7-VP1u bioconjugates by centrifugation through 20% sucrose (150,000× *g*, 2 h, 4 °C). The final PP7-VP1u stock concentration was quantified by PCR and SDS-PAGE.

### 2.5. SDS-PAGE and Western blot

SDS-PAGE was performed to analyze the purity of the expressed VP1u variants and to verify the efficiency of the PP7-VP1u coupling. For Western blot, the proteins in the gel were blotted on a PVDF membrane and blocked overnight with 5% milk. For detection, a rat anti-FLAG antibody was used, followed by an HPR-coupled secondary antibody. Detection was performed with SuperSignal West Dura Extended Duration Substrate (Thermo Fisher, Waltham, MA, USA) and visualized with a photo film.

### 2.6. Detection of PP7-VP1u and B19V Uptake by qPCR and Confocal Microscopy

The different cell types (3 × 10^5^) were resuspended in 200 µL PBS and incubated with 10^10^ PP7-VP1u bioconjugates (B19V or simian) or B19V from human plasma (10^10^) at 37 °C for 30 min. Subsequently, cells were washed twice with 1 mL PBS, trypsinized for 4 min (0.25% trypsin/EDTA solution (Sigma, St. Louis, MO, USA) diluted to 50% in PBS) at 37 °C, and washed twice more with 1 mL PBS. For PP7-VP1u quantification, cells were lysed in 140 μL lysis buffer (0.5% Tween-20 in PBS). The cell debris was removed by centrifugation, and the supernatant was transferred to a new tube. The phage RNA of the internalized bioconjugates was extracted with a QlAamp Viral RNA Mini Kit (Qiagen, Hilden, Germany) and quantified by RT-PCR with iTaq SybrGreen qPCR (BioRad, Hercules, CA, USA) and PP7-specific primers; forward, 5′-GGC AAC TGA GCA TAA CGG CAC-3′; reverse, 5′-GCT CCA TAG CGA TGA AGC GAA C-3′. For B19V quantification, intracellular B19V genomes were extracted from the cell pellet with a DNeasy Blood and Tissue Kit (Qiagen) and quantified by PCR with iTaq SybrGreen qPCR (BioRad) and B19V-specific primers; forward (nt 413-433), 5′-GGG CAG CCA TTT TAA GTG TTT-3′; reverse (nt 534-552), 5′-CCA GGA AAA AGC AGC CCA G-3′. Results were normalized by quantification of the β-actin gene and expressed as a percentage of maximal uptake.

To visualize internalized virus, the infected cells were resuspended in 20 μL PBS, spotted on coverslips, fixed with acetone/methanol (1:1) at −20 °C for 4 min, and allowed to dry. Samples incubated with Atto488-labeled PP7-VP1u bioconjugates were directly mounted with Mowiol containing DAPI. Immunofluorescence detection of B19V was carried out with MAb 860-55D (Mikrogen, Neuried, Germany) against intact capsids, followed by a goat anti-human Alexa Fluor 549 (Agilent Technologies, Santa Clara, CA, USA). Internalized viruses were visualized by confocal microscopy (LSM 880, Zeiss, Jena, Germany) using a 63x oil-immersion objective.

### 2.7. Detection of Recombinant VP1u in Cells by Confocal Microscopy

Recombinant VP1u constructs (50 ng) were incubated with a rat anti-FLAG antibody for 1 h at 37 °C. Subsequently, the VP1u constructs were incubated with cells (3 × 10^5^) in 200 µL PBS at 4 °C for 40 min to allow binding or at 37 °C to allow binding and internalization. Cells were processed for immunofluorescence as described above. Detection of recombinant VP1u constructs was carried out with a secondary goat anti-rat antibody, Alexa Fluor 488, or a goat anti-rabbit antibody, Alexa Fluor 594 (Agilent Technologies, Santa Clara, CA, USA), and visualized by confocal microscopy.

### 2.8. Competition and Neutralization Assays

The internalization of PP7-VP1u constructs and native B19V were examined in the presence of recombinant VP1u or PP7-VP1u. Cells were resuspended in 200 µL PBS and incubated with 500 ng of recombinant VP1u or a 10x molar excess of PP7-VP1u for 40 min at 4 °C prior to incubation with PP7-VP1u or B19V at 37 °C for 30 min. Subsequently, samples were processed for qPCR and/or immunofluorescence microscopy as specified above.

The capacity of a human antibody against the N-VP1u of B19V (MAb 1418-1; aa 30 to 42) [37,38] to recognize and block the uptake of human and simian VP1u was examined. PP7-VP1u bioconjugates were incubated with a 200x molar excess of MAb 1418-1 for 1 h at 4 °C. After incubation, PP7-VP1u uptake was examined in UT7/Epo cells by RT-qPCR as described above.

### 2.9. In Silico Predictions

VP1u sequence alignments were carried out with the Clustal Omega multiple sequence alignment program [39]. Ab initio tertiary structure prediction of the RBDs was performed using the QUARK server (https://zhanggroup.org/QUARK/, accessed on 12 September 2020) [40]. The structures were visualized and analyzed with PyMOL molecular graphics software (version 2.5.0).

## 3. Results

### 3.1. PP7-VP1u Bioconjugation

PP7-VP1u bioconjugates were assembled by bio-orthogonal click chemistry (Figure 1A). PP7 was coupled to the heterobifunctional crosslinker TCO-NHS, and VP1u was crosslinked to the heterobifunctional crosslinker MeTz-Mal. Subsequently, PP7-TCO and VP1u-MeTz were crosslinked by inverse-electron-demand Diels–Alder reaction. SDS-PAGE was performed to verify the purity of PP7, PP7-TCO, and PP7-VP1u. The PP7 bacteriophage capsid protein is visible at 13.9 kDa. Additionally, the capsid-associated maturation protein is slightly visible at 50.8 kDa (Figure 1B). Sensitive Western blot analysis showed that approximately three VP1u units were crosslinked to one PP7 capsid and that non-coupled VP1u was effectively removed after the crosslinking reaction (Figure 1C).

### 3.2. PP7-VP1u Is Internalized into UT7/Epo Cells and Competes with B19V

The capacity of the PP7-VP1u bioconjugate to bind and internalize UT7/Epo cells was tested by RT-qPCR. Following incubation for 30 min at 37 °C, the cells were briefly trypsinized to remove non-internalized PP7-VP1u particles. The sample incubated at 4°C served as a control of the trypsinization treatment. Internalized PP7-VP1u constructs were quantified by RT-PCR. The result showed that only the VP1u-bioconjugated construct incubated at 37 °C was able to internalize. Competition with recombinant VP1u efficiently inhibited PP7-VP1u uptake, confirming that the construct internalizes following interaction with VP1uR (Figure 2A). When added to the cells before native B19V, the PP7-VP1u particles interfered significantly with B19V uptake, although less efficiently than the recombinant VP1u (Figure 2B). The internalization kinetics of B19V and PP7-VP1u were examined in parallel at 5 min intervals. The results revealed a similar uptake profile between the native virus and the bioconjugate phage capsid (Figure 2C).

The capacity of PP7-VP1u to internalize UT7/Epo cells was also tested by fluorescence microscopy. Following incubation for 30 min at 37 °C, the cells were washed, trypsinized to remove non-internalized particles, fixed, and examined by fluorescence microscopy. B19V- and PP7-TCO-infected cells were used as a positive and negative control, respectively. At 37 °C, PP7-VP1u particles appear with the typical clustered intracellular distribution similar to that observed in cells infected with B19V. Binding but not uptake is observed at 4 °C (Figure 2D). Taken together, PP7-VP1u internalized into UT7/Epo cells with comparable efficiency as native B19V, confirming its suitability as a sensitive and quantitative VP1uR marker.

### 3.3. VP1uR Expression is Restricted to Epo-Dependent Erythroid Cells

The capacity of PP7-VP1u to bind and internalize UT7/Epo cells with similar efficiency as B19V validates the use of the phage bioconjugate as a sensitive marker for the quantitative determination of VP1uR expression. We first examined the expression of VP1uR in non-hematopoietic cells. To this end, PP7-VP1u was incubated with cells from the lung (A549 and MRC-5), kidney (HEK 293T and NB324), liver (HepG2), and cervix (HeLa). In contrast to UT7/Epo cells, which served as a positive control, no significant signal of internalized PP7 RNA or capsid fluorescence signal was detected in any non-hematopoietic cells tested, confirming the restricted expression profile of VP1uR (Figure 3A,B).

We next tested cell lines originating from distinct differentiation stages of the hematopoietic hierarchy roadmap, i.e., lymphopoiesis (REH cells), granulo- and monocytopoiesis (KG1a and Ku812Ep6), erythropoiesis (UT7/Epo), and multipotent hematopoietic stem cell (K562). For comparability, the internalization assay with PP7-VP1u was performed in the same way as described above for non-hematopoietic cells. As quantified by RT-PCR (Figure 3C), only cells involved in Epo-dependent erythroid differentiation, Ku812Ep6 and UT7/Epo, expressed VP1uR abundantly. Whereas K562 had a low but detectable expression, reaching approximately 20% of that observed in UT7/Epo cells, KG1a and lymphoid REH cells did not show a significant expression of VP1uR. The corresponding fluorescence microscopy images confirm the predominant VP1uR expression in the Epo-dependent erythroid cell lines with a clear intracellular signal, a weak signal in K562, a weak pericellular signal in KG1a cells, and no signal in REH cells (Figure 3D).

### 3.4. Quantification of VP1uR Expression at Progressive Erythroid Differentiation Stages

To accurately resolve the expression profile of VP1uR and erythroid differentiation, CD34+ hematopoietic stem cells were cultured in erythroid differentiation medium containing Epo and tested daily for PP7-VP1u uptake. The results show that after induction of erythroid differentiation, VP1uR expression increases until day 8 and decreases thereafter (Figure 4A). The quantitative data are supported by the fluorescence microscopy images, which show PP7-VP1u internalization after 3 and 8 days of differentiation. Immunostaining with a glycophorin A antibody confirmed the commitment and differentiation of the cells toward the erythroid lineage (Figure 4B). These findings confirm that VP1uR expression is upregulated in EPCs upon Epo stimulation.

### 3.5. VP1uR Expression Can Be Induced by Cellular Reprogramming towards Erythroid lineage

HuDEP and HiDEP cells were established from umbilical cord blood CD34+ cells and iPS derived from human fibroblasts, respectively (Figure 4C). These cells express erythroid-specific markers, which are upregulated after the induction of differentiation [41]. Neither CD34+ blood cells [26] nor human fibroblast cells (Figure 3A,B), used as precursor cells, express VP1uR. In clear contrast, VP1uR expression was detected following cellular reprogramming towards erythroid lineage (Figure 4D). The quantitative RT-PCR result was confirmed by fluorescence microscopy with the PP7-VP1u construct (Figure 4E).

### 3.6. Structural Comparison of the RBD of B19V and Related Primate Erythroparvoviruses

The receptor-binding domain (RBD) in the N-terminal region of VP1u of B19V was characterized in detail in our previous studies. It was shown that the B19V RBD has a well-defined structure of a three-helix fold, which forms a spatial cluster of internalization-important amino acids at the interface of helices 1 and 3 [27]. Considering the importance of the N-VP1u RBD in B19V tropism and the infection similarities between B19V and related primate erythroparvoviruses, we sought to compare the sequence and predicted structure of their N-VP1u.

As shown in Figure 5A, the primate erythroparvoviruses cluster in a monophyletic group within the genus Erythroparvovirus. Multiple sequence alignment shows significant differences in the N-VP1u amino acid sequence from primate erythroparvoviruses (Figure 5B). Only a few amino acids are conserved in the RBD of B19V, SPV, RhMPV, and PtMPV. Interestingly, amino acids F25 and L59, which were characterized as internalization-relevant for B19V VP1u [27], are also conserved in the other primate erythroparvoviruses. Further internalization-relevant amino acids of B19V VP1u are not conserved by identical amino acids, but they are replaced by amino acids with comparable properties. For example, in helix 1, the polar amino acid glutamine is conserved in B19V (Q22) and PtMPV (Q25), whereas it is replaced by glutamate in SPV (E25) and RhMPV (E27). Similarly, internalization-important hydrophobic amino acids of B19V are replaced by analogous hydrophobic amino acids within other primate erythroparvoviruses.

Secondary and tertiary protein structure predictions of N-VP1u were performed with the QUARK server [40]. Although the sequences are barely conserved, the predicted helix distribution and the tertiary structure of N-VP1u of primate erythroparvoviruses show striking similarities (Figure 5C). The RBDs of human B19V and PtMPV is composed of three helices. Both RBDs show an identical helix distribution, leading to a very similar 3D structure. The three-helix fold forms a spatial cluster of important amino acids at the interface of helices 1 and 3. The RBDs of SPV and RhMPV harbors four helices, both folding to a virtually identical 3D structure. The additional helix is located between helices 2 and 3, encoded by an additional sequence that is not present in the sequence of B19V and PtMPV. The four-helix fold builds a spatial cluster of important amino acids that are located on the interface of helices 1 and 3 (Figure 5D). Taken together, the internalization-important amino acids show a remarkably conserved structural arrangement within the RBDs of all primate erythroparvoviruses, and the phylogenetic relationship between the viruses correlates well with the similarity in their predicted RBD models.

### 3.7. N-VP1u of SPV Harbors a Functional RBD, Mediating Virus Uptake into Erythroid Cells

The striking similarity observed between the predicted N-VP1u structure of B19V and that of other primate erythroparvoviruses suggests a common RBD function. To verify the presence of a functional RBD in the VP1u of non-human primate erythroparvoviruses, we generated a simian PP7-VP1u bioconjugate following the same approach as that used for B19V (Figure 1). Incubation of the simian PP7-VP1u bioconjugate with non-hematopoietic (HeLa), hematopoietic (KG1a), and erythropoietic (UT7/Epo) cell lines at 37 °C for 1h showed strong internalization of simian PP7-VP1u into UT7/Epo but not into HeLa or KG1a cells (Figure 6A,B). This finding confirms the presence of a functional RBD in the N-VP1u of SPV that is able to mediate virus uptake into human erythroid cells.

Furthermore, the uptake of simian PP7-VP1u was tested in competition with recombinant VP1u of SPV and B19V. Whereas simian VP1u was able to efficiently block internalization of the bioconjugate, the inhibitory effect of B19V VP1u on simian PP7-VP1u was not significant (Figure 6C). On the other hand, simian VP1u was able to disturb the internalization of native B19V but not as efficiently as the B19 VP1u (Figure 6D). These results suggest that SPV and B19V binding to VP1uR may involve neighboring domains of the receptor, resulting in a partial competition between the two species.

During B19V viremia, antibodies produced against RBDs in N-VP1u are crucial to clearing the infection. An antibody targeting N-VP1u (aa 30 to 42; MAb 1418-1) obtained from a B19V-infected patient was shown to efficiently neutralize B19V infection [37,38]. Considering the zoonotic potential of SPV, we tested the capacity of MAb 1418-1 to block simian PP7-VP1u uptake in UT7/Epo cells. Whereas the antibody effectively inhibited B19 PP7-VP1u uptake, simian PP7-VP1u internalization remained undisturbed (Figure 6E), suggesting that the differences in structure and sequence within the RBD allow SPV to evade recognition by human antibodies.

## 4. Discussion

### 4.1. Expression Profile of VP1uR

The N terminal of VP1u of B19V harbors an RBD required for virus uptake [25,27]. The VP1u cognate receptor, herein named VP1uR, has not yet been identified, but its expression profile seems restricted to the few cell types that B19V can infect, i.e., EPCs at Epo-dependent differentiation stages and the erythroid cell lines UT7/Epo and Ku812Ep6 [26,36]. Considering its central role in infection, a detailed characterization of the expression profile of VP1uR is needed to better understand the tropism and pathogenesis of B19V.

In this study, we developed a PP7 bacteriophage-VP1u bioconjugate as a sensitive and quantitative marker to analyze VP1uR expression by fluorescence microscopy and RT-qPCR. The PP7-VP1u construct allowed for the study of VP1u-dependent uptake without the involvement of other interacting domains within the B19V capsid. The PP7 capsid has a similar size as that of B19V, does not bind to eukaryotic cells, and contains a specific RNA genome that was used for quantification. Functional assays confirmed that the PP7-VP1u construct can bind and internalize erythroid cells with similar efficiency as native B19V (Figure 2C,D). Competition assays with recombinant VP1u showed that PP7-VP1u internalization is mediated exclusively by VP1u and its cognate receptor, without the engagement of additional receptor molecules (Figure 2A,B). Accordingly, the PP7-VP1u bioconjugate appeared as a sensitive method to detect and quantify VP1uR expression.

VP1uR expression was systematically investigated in hematopoietic and non-hematopoietic cells derived from different tissues. The functional receptor was exclusively detected in the erythroid cell lines UT7/Epo and Ku812Ep6 (Figure 3), in EPCs at Epo-dependent differentiation stages, and in cells reprogrammed towards the erythroid lineage (Figure 4). These results confirm that VP1uR expression is tightly linked to erythropoiesis, determining the marked erythroid tropism of B19V. VP1uR expression was significantly increased in Ku812Ep6 cells compared to the reference cell line, UT7/Epo. Ku812 cells were established from a patient with a blastic crisis of chronic myelogenous leukemia [44]. Differentiation toward the erythroid lineage in the presence of Epo led to the clone Ku812Ep6, which displayed a significantly increased susceptibility to B19V [45]. Epo-independent hematopoietic cell lines, such as KG1a and K562, show a low but detectable VP1uR expression (Figure 3C,D), suggesting that Epo per se does not induce VP1uR expression but rather supports the survival and proliferation of the erythroid cell population that expresses VP1uR.

The stimulation of isolated CD34+ cells with erythropoiesis-supporting cytokines triggered erythroid differentiation, which was evidenced by the expression of glycophorin A (GPA). VP1uR expression was upregulated during the first days of differentiation, reaching the highest level after eight days, and downregulated during the terminal erythroid differentiation stages (Figure 4). These results confirm previous studies wherein a direct correlation between B19V uptake and erythroid differentiation was observed [26,46].

VP1uR expression was detected in cells reprogrammed towards the erythroid lineage. HuDEP and HiDEP cells are immortalized EPCs established from umbilical cord blood CD34+ cells and induced pluripotent stem (iPS) cells (derived from fibroblasts) expressing TAL1, respectively [41]. TAL1 is essential in early hematopoiesis and erythroid differentiation [47,48]. Immortalization was achieved by the induction and expression of HPV16-E6/E7. HuDEP and HiDEP cells express the erythroid-specific marker GPA, whereas CD36 and c-KIT (CD117), which are markers of immature erythroid cells, are detected at very low levels [41]. PP7-VP1u was able to internalize HuDEP and HiDEP cells, although to a lesser extent compared to UT7/Epo cells. These cells exhibit gene expression patterns characteristic of terminal erythroid differentiation stages [49], which correspond with the observed downregulation of VP1uR expression during erythroblast stages.

### 4.2. Characterization of RBDs from Non-Human Primate Erythroparvoviruses

B19V is closely related to three other primate erythroparvoviruses, i.e., SPV, RhMPV, and PtMPV [29,30,32]. The tropisms and clinical features observed in natural and experimental infections with these viruses resemble those of B19V and suggest their possible zoonotic potential [31,32,33,34,35]. Considering that the marked erythroid tropism of B19V is mediated by the RBD in N-VP1u, we hypothesized that a similar RBD might be present in N-VP1u from other primate erythroparvoviruses.

Previous investigations revealed that the B19V RBD has a well-defined structure of a three-helix fold, which forms a spatial cluster of internalization-relevant amino acids at the interface of helices 1 and 3 [27]. We compared the primary sequences located in the putative RBD within the VP1u region, and modulated in silico the RBD with the computer algorithm QUARK (Figure 5). The predicted RBD model of PtMPV exhibited a remarkable resemblance to that of B19V. The SPV and RhMPV RBD differed from the B19V RBD by an additional α-helix between helices 2 and 3. Although the primary amino acid sequences of primate VP1u regions were moderately conserved, the in silico modulations exhibited a similar spatial arrangement of helices 1 and 3. Internalization-important amino acids that were previously identified for B19V [27] were conserved by either identical amino acids or conservative replacements, resulting in a comparable spatial arrangement for all primate erythroparvoviruses.

The remarkable similarity of the predicted N-VP1u structure from the different primate erythroparvoviruses suggests a common function as an RBD required for virus uptake into erythroid progenitor cells. In line with this hypothesis, simian VP1u was able to specifically internalize into human UT7/Epo cells (Figure 6A,B), confirming its function as an RBD for virus uptake into erythroid cells. The close homology of all four primate erythroparvoviruses and the similar structure and function of their RBDs suggest that the uptake mechanism of these viruses is evolutionarily conserved. One possible evolutionary scenario is the infection of humans by an ancestor of SPV and the deletion of the additional helix in the RBD during the adaptive course of evolution, which resulted in the characteristic three-helix cluster of B19V. However, it is also conceivable that PtMPV represents an intermediate evolutionary step between SPV and B19V. Taken together, primate erythroparvoviruses could have a zoonotic potential for humans and be of particular concern for immunocompromised persons or pregnant women. Nevertheless, the results of this study are limited to the step of virus attachment and internalization and do not allow for conclusions about the permissiveness of human erythroid cells to non-human primate erythroparvoviruses.

The question remains whether B19V and the related non-human primate viruses rely on the same receptor structure for internalization. The fact that B19V VP1u was not able to interfere with simian PP7-VP1u uptake and simian VP1u only partially competed with B19V internalization (Figure 6C,D) suggests that B19V and SPV interaction with VP1uR may involve different binding sites. The possibility of two unrelated receptors, however, appears rather unlikely, since the RBD of SPV and B19V showed a similar structure with a comparable spatial cluster of internalization-important amino acids, as well as the same extraordinary specificity for erythroid cell types. B19V and SPV may recognize distinctive surface glycosylation patterns of the same receptor, where the steric hindrance of the additional helix in the RBD of SPV could play a key role. Finally, the internalization of simian VP1u was not affected in presence of MAb 1418-1 (Figure 6E). The differences in the structure and sequence of the RBD of SPV allow the simian virus to escape the neutralizing human antibody while maintaining the capacity to recognize VP1uR. Accordingly, SPV VP1u could be exploited for the therapeutic targeting of human erythroid cells without the limitation of pre-existing B19V-neutralizing antibodies.

## 5. Conclusions

The RBD in the N-VP1u of B19V interacts with VP1uR for virus internalization into susceptible cells. VP1uR expression was systematically analyzed in cells from different tissues by using a specific and sensitive marker based on a phage-VP1u bioconjugate. The receptor was exclusively detected in cells of the erythroid lineage, explaining the marked erythroid tropism of B19V. Structural and functional studies of N-VP1u from related non-human primate erythroparvoviruses revealed the presence of an analogous RBD-mediating virus internalization into human erythroid cells. These findings underline the close evolutionary relationship among human and animal erythroparvoviruses and further substantiate their zoonotic potential.

## Figures and Tables

**Figure 1 viruses-14-00420-f001:**
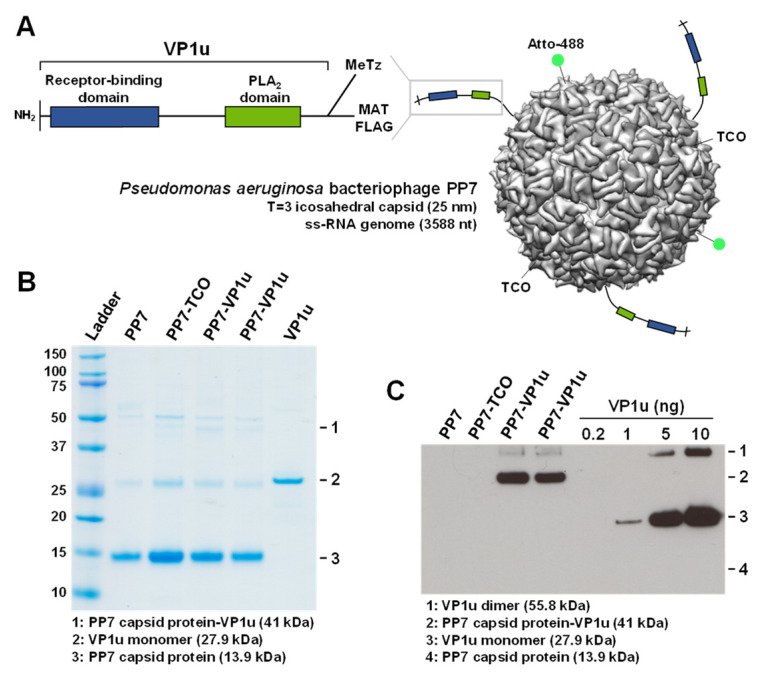
PP7-VP1u bioconjugation. (**A**) Schematic depiction of the PP7-VP1u construct, consisting of a bacteriophage PP7 capsid and three VP1u units of B19V. VP1u contains a C-terminal MAT tag for purification, a C-terminal FLAG tag for detection, and a unique cysteine for crosslinking. Additionally, the capsid was labeled with NHS-Atto488. (**B**) SDS-PAGE and (**C**) Western blot analysis of PP7 capsids, recombinant VP1u, and PP7-VP1u construct (two replicates). VP1u and PP7-VP1u conjugates were detected with an anti-FLAG antibody.

**Figure 2 viruses-14-00420-f002:**
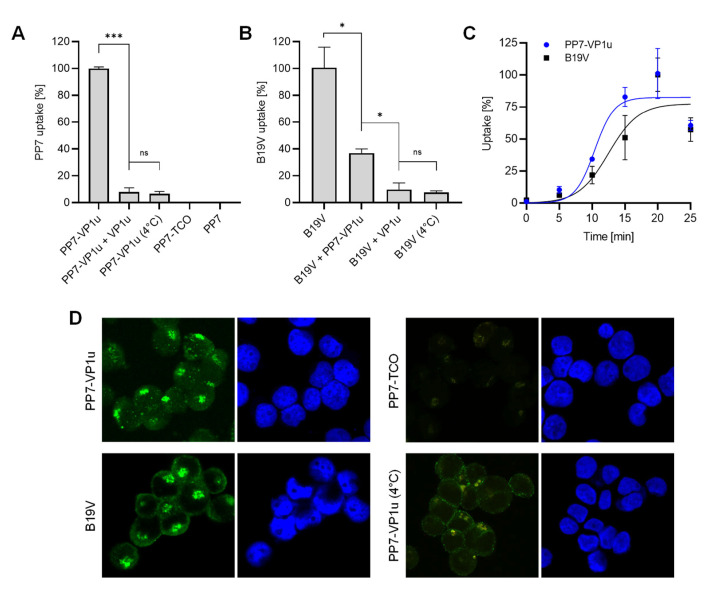
PP7-VP1u uptake into UT7/Epo cells. (**A**) PP7-VP1u uptake was quantified by RT-qPCR. For the competition, recombinant VP1u (500 ng) was added to the cells at 4 °C 40 min before incubation with PP7-VP1u. (**B**) Internalization of native B19V into UT7/Epo cells was quantified by PCR. For the competition, PP7-VP1u (10x molecular excess) or recombinant VP1u (500 ng) was added to the cells at 4 °C for 40 min before infection with B19V. The sample at 4 °C served as a negative control (no internalization). (**C**) Internalization kinetics determined by RT-qPCR (PP7-VP1u) or qPCR (B19V) at 5 min intervals. (**D**) Binding and uptake of Atto488-labeled PP7-VP1u constructs and native B19V. B19V was detected with human MAb 860-55D against capsids and stained with secondary Alexa-Fluor 488 anti-human antibody. PP7-TCO served as a negative control. The quantitative RT-PCR results are presented as the mean ± SD of three independent experiments. *** *p* < 0.001; * *p* < 0.05; ns, not significant.

**Figure 3 viruses-14-00420-f003:**
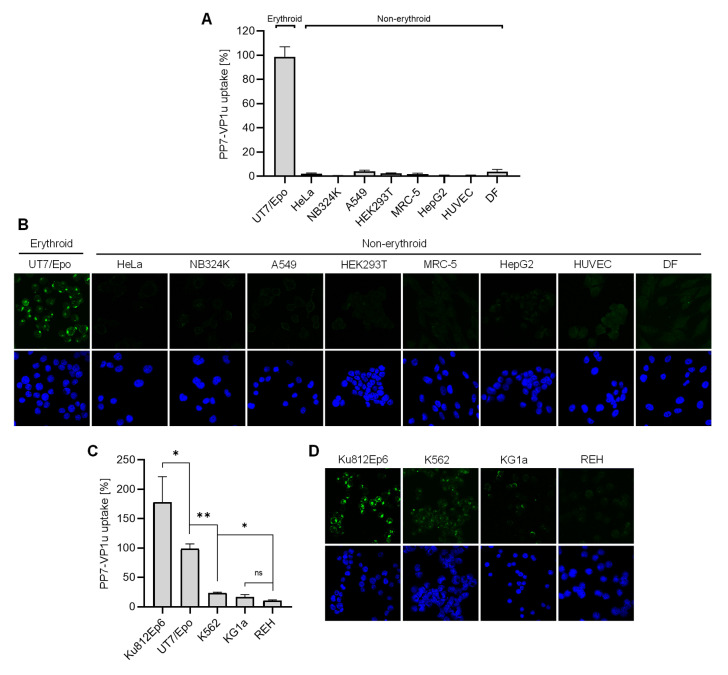
Expression profile of VP1uR in erythroid and non-erythroid cells. PP7-VP1u was incubated with different human cell types derived from non-erythroid tissues for 30 min at 37 °C. UT7/Epo served as a positive control. The internalized constructs were quantified by RT-PCR (**A**) and detected by fluorescence confocal microscopy (**B**). PP7-VP1u was incubated with different human hematopoietic cell lines for 30 min at 37 °C. UT7/Epo served as a positive control. The internalized constructs were quantified by RT-PCR (**C**) and detected by fluorescence confocal microscopy (**D**). All samples were treated with trypsin before RNA extraction to remove bound but not internalized constructs. The quantitative RT-PCR results are presented as the mean ± SD of three independent experiments. ** *p* < 0.01; * *p* < 0.05.

**Figure 4 viruses-14-00420-f004:**
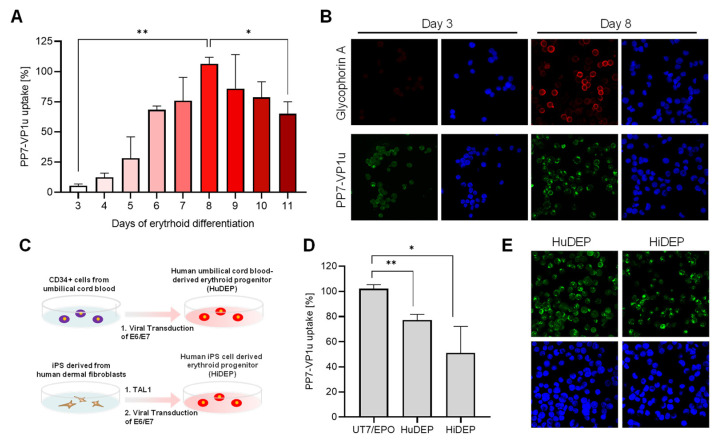
VP1uR expression is upregulated during erythroid differentiation. (**A**) EPCs at progressive differentiation phases were incubated with PP7-VP1u for 30 min at 37 °C and quantified by RT-PCR. (**B**) Confocal microscopy images of EPCs at days 3 and 8 of erythroid differentiation. Cells were incubated with PP7-VP1u or immunostained with a glycophorin A antibody. (**C**) Schematic depiction of the erythroid reprogramming strategy [41]. (**D**) PP7-VP1u bioconjugates were incubated with HuDEP and HiDEP cells for 30 min at 37 °C. Internalization was quantified by RT-qPCR and compared with UT7/Epo cells. (**E**) Representative images of HuDEP and HiDEP cells incubated with PP7-VP1u and visualized by confocal microscopy. The quantitative RT-PCR results are presented as the mean ± SD of two (**A**) or three (**D**) independent experiments. ** *p* < 0.01; * *p* < 0.05.

**Figure 5 viruses-14-00420-f005:**
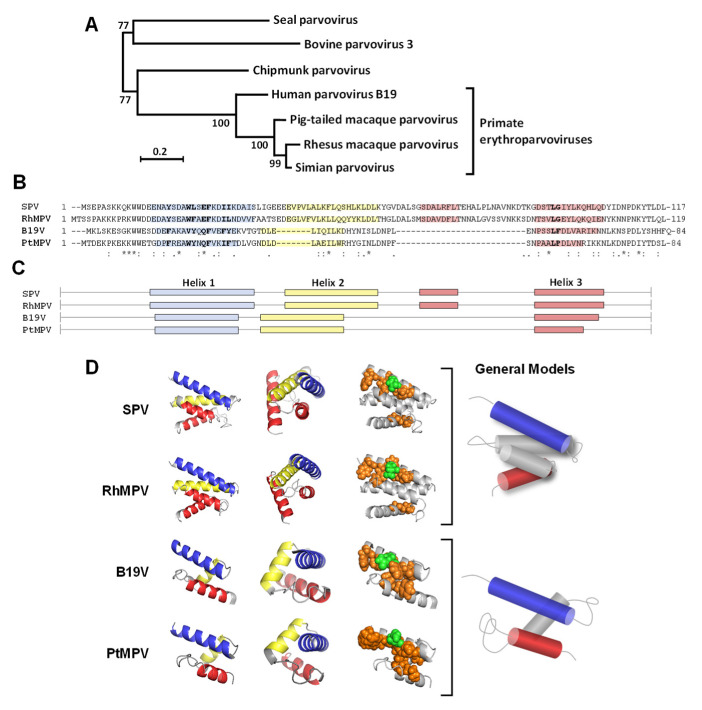
N-VP1u from primate erythroparvoviruses. (**A**) Phylogenetic tree of the genus Erythroparvovirus. VP1 sequences were aligned with MUSCLE configured for the highest accuracy [42], and the tree was reconstructed using the maximum-likelihood method implemented in the PhyML program [43]. Bootstrap values are indicated. (**B**) N-VP1u sequences were aligned using the Clustal Omega multiple sequence alignment program. Relevant amino acids for VP1u internalization are highlighted in bold letters. (**C**) Helix distribution according to secondary structure prediction in QUARK modeling. Helix 1 is blue, helix 2 is yellow, and helix 3 is red. (**D**) The RBDs (SPV AA 14-117, RhMPV AA 15-119, B19V 14-84, and PtMPV AA 14-84) were modeled by QUARK and visualized with PyMOL, showing similar high-confidence structure predictions of the different viruses. First column: front view with the N-terminus top left and the C-terminus in the lower part; second column: side view of the models; third column: spatial cluster of the important amino acids as spheres in the helical structure (green: polar; orange: hydrophobic). A schematic representation of the two models of helix configuration is shown on the right. SPV: simian parvovirus; RhMPV; rhesus macaque parvovirus; B19V: parvovirus B19; PtMPV: pig-tailed macaque parvovirus.

**Figure 6 viruses-14-00420-f006:**
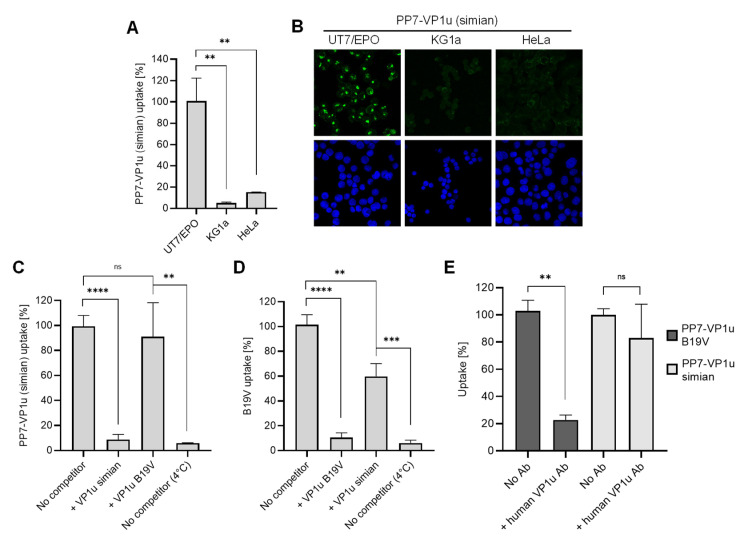
Uptake of simian PP7-VP1u into human cells. Simian PP7-VP1u was incubated with UT7/Epo, KG1a, and HeLa cells for 30 min at 37 °C. The internalized constructs were quantified by RT-PCR (**A**) and detected by fluorescence confocal microscopy (**B**). (**C**,**D**) Internalization of simian PP7-VP1u and native B19V into UT7/Epo cells in the presence of competitors. Samples incubated without competitors at 37 °C and 4 °C served as a reference for maximal internalization and complete block, respectively. (**E**) B19V and simian PP7-VP1u internalization into UT7/Epo cells was carried out in the absence or presence of a 200-fold molecular excess of MAb 1418-1 against B19V VP1u. Internalized bioconjugates were quantified by RT-PCR. The quantitative RT-PCR results are presented as the mean ± SD of three independent experiments. **** *p* < 0.0001; *** *p* < 0.001; ** *p* < 0.01; ns, not significant.

## Data Availability

The data presented in this study are available on request from the corresponding author.

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
