# Peer review of "A Conserved Receptor-Binding Domain in the VP1u of Primate Erythroparvoviruses Determines the Marked Tropism for Erythroid Cells"

_viruses, 2022, doi:10.3390/v14020420_

Round 1

Reviewer 1 Report

A well prepared and presented study of the role of the VP1 unique region of B19 and related viruses in their tropism for cells of the erythroid lineage. The data here is based on expressing the protein as a separate protein, or as a fusion with a PP7 phage system. In either case the protein bound to the differentiated cells, and there was a similarity in the results of the human virus and those that infect primates. Overall I have no comments about this as the work appears to be well conducted and clearly presented.

The identity of the receptor is a major question, and it is not clear why that has not yet been identified (or what its properties are - is it a protein or glycan?) The connection to the previously identified P antigen (globoside) is not clearly discussed, but presumably that is not involved.

Is there another receptor that binds the capsid protein, like the well characterized receptors of other parvoviruses - that is not mentioned as far as I can tell.

Author Response

A well prepared and presented study of the role of the VP1 unique region of B19 and related viruses in their tropism for cells of the erythroid lineage. The data here is based on expressing the protein as a separate protein, or as a fusion with a PP7 phage system. In either case the protein bound to the differentiated cells, and there was a similarity in the results of the human virus and those that infect primates. Overall I have no comments about this as the work appears to be well conducted and clearly presented.

The identity of the receptor is a major question, and it is not clear why that has not yet been identified (or what its properties are - is it a protein or glycan?) The connection to the previously identified P antigen (globoside) is not clearly discussed, but presumably that is not involved.

The identification of the VP1uR is in progress. The VP1uR and globoside are two distinct structures, although both are involved in the infection.

Is there another receptor that binds the capsid protein, like the well characterized receptors of other parvoviruses - that is not mentioned as far as I can tell.

Ku80 and integrin α5β1 were shown to bind B19V. They may participate in the uptake process in certain cells.

Reviewer 2 Report

In this paper entitled "A conserved receptor-binding domain in the VP1u of Primate erythroparvoviruses determines the marked tropism for erythroid cells", the authors aimed to analyze the ability of VP1u, a unique domain present in the erythroparvoviruses structural protein VP1, to restrict tropism to erythroid cells. For this purpose, they made use of a chimeric construct composed of PP7 particles harboring 3 VP1u RBD sequences, named PP7-VP1u. With this construct, the authors shows a strict tropism to erythroid cells and a competition with recombinant VP1u and native B19V internalization.  In addition, by using a similar construct with simian VP1u (PP7-VP1u simian), they showed its internalization in human erythroid cells. Altogether, the authors conclude to a common mechanism of internalization between primates, underlying a close evolution trait of erythroparvoviruses.

Overall, this paper presents an interesting technique to analyze VP1u mediated internalization with no need of change of conformation, thus providing specific results regarding VP1u role in B19V internalization. However some conclusions linked to VP1u receptor expression are overstated as VP1u receptor has not yet been clearly defined. Thus results showed indirectly its existence on cells, and data could not be sustained by parallel analysis of VP1u receptor expression invalidation of the proper VP1u receptor. The author should highlight this point clearly in the paper and be caution with the mention of VP1uR expression as the demonstration is indirect.

Moreover, no bibliographic mention is noticed about the internalization and latency of B19V in infected humans, leading to inflammation phenotype of non-erythroid organs and tissues. It would have been thus interesting to refer to these publications and to analyze PP7VP1u internalization in brain, heart or mesenchymal cells. 

Specific comments:

1) Protocols are not deeply enough addressed to properly understand the scope and the significance of the data presented: the calculation of the uptake percentage in figures 2, 3, 4 and 6 is blur, and the means of 100% uptake is not clearly explained. How many copies are internalized per cell? Also in material and methods, line 191, the quantity of bioconjugates and B19V is not relevant as there is not units mentioned.

2) In figure 2, recombinant VP1u is used to compete with PP7-VP1u internalization. It would have been interesting to apply increasing range of recombinant VP1u to properly quantify the affinity of PP7-VP1u to the VP1u receptor. 

For competition experiments, VP1u is added 40 minutes before PP7-VP1u. Is it known if recombinant VP1u is internalized or did it binds to its receptor in an irreversible manner? This question should be discussed by the authors. A similar question is raised by figure 2C with 100% uptake at 15 minutes. 40 minutes after a first application of PP7-VP1U, are cells able to internalize a new batch of PP7-VP1U? 

3) Figure 6 showed the use of simian PP7-VP1u on human erythroid cells. It would have been interesting to show the complementary experiment using human PP7-VP1u on simian, rhesus and pig-tailed macaque cells. 

Minor comments:

  • Lines 19 & 22: what is the mean of N-VP1u?
  • Line 26: Parvovirus B19 and B19V are two redundant keywords
  • Line 43: B19V entry route is not only respiratory. Please clarify.
  • Line 45: infected EPC leads to erythroid disorders
  • Line 109: precise the UT7/Epo cell line used in the study
  • paragraph 2.1: no mention of NB324K cells used in the paper
  • Line 124: Please clarify HSA
  • Line 138: correct trough by through
  • Line 192: precise trypsination condition as it is a key point of internalization analysis
  • Line 204-210: Immunofluorescence protocol should be precisely described. Please clarify fixation method as cold acetone/methanol. What are the permeabilisation steps to allow detection of internalized VP1 or B19V? The blocking steps? 
  • Line 197 and all along the paper, PCR analysis (RT-PCR for PP7) should be corrected as qPCR (RT-qPCR)
  • Lines 201-202: position of the targeted region of B19V primers? 
  • Figure 2D, 3B, 3D, 4B, 4E, 6B: scale bars are missing. 
  • Figure 4 D: statistical analysis of 2 independent experiments is not relevant. 
  • Line 338: E instead of D

Author Response

In this paper entitled "A conserved receptor-binding domain in the VP1u of Primate erythroparvoviruses determines the marked tropism for erythroid cells", the authors aimed to analyze the ability of VP1u, a unique domain present in the erythroparvoviruses structural protein VP1, to restrict tropism to erythroid cells. For this purpose, they made use of a chimeric construct composed of PP7 particles harboring 3 VP1u RBD sequences, named PP7-VP1u. With this construct, the authors shows a strict tropism to erythroid cells and a competition with recombinant VP1u and native B19V internalization.  In addition, by using a similar construct with simian VP1u (PP7-VP1u simian), they showed its internalization in human erythroid cells. Altogether, the authors conclude to a common mechanism of internalization between primates, underlying a close evolution trait of erythroparvoviruses.

Overall, this paper presents an interesting technique to analyze VP1u mediated internalization with no need of change of conformation, thus providing specific results regarding VP1u role in B19V internalization. However some conclusions linked to VP1u receptor expression are overstated as VP1u receptor has not yet been clearly defined. Thus results showed indirectly its existence on cells, and data could not be sustained by parallel analysis of VP1u receptor expression invalidation of the proper VP1u receptor. The author should highlight this point clearly in the paper and be caution with the mention of VP1uR expression as the demonstration is indirect.

To detect and quantify viral receptors expressed on the plasma membrane of cells, specific antibodies are typically used. This is only possible when the identity of the receptor is known. The receptor-binding domain (RBD) construct used in this study shares with specific antibodies the ability to bind the receptor. However, the RBD construct has additional advantages over specific antibodies, i.e., does not only bind the VP1uR, but provides information about function (virus uptake), binds the specific site in the receptor recognized by the virus (antibodies may bind unrelated sites in the receptor or distinct isoforms), and allow accurate quantification of the expression level, as we can use quantitative RT-PCR.

In the manuscript we mentioned the advantage of using a functional viral RBD construct:

L104-106 “This approach allowed for a more sensitive and quantitative determination of the expression profile of VP1uR in multiple cells types and the possibility to study VP1uR-dependent virus uptake by RT-qPCR”.

Moreover, no bibliographic mention is noticed about the internalization and latency of B19V in infected humans, leading to inflammation phenotype of non-erythroid organs and tissues. It would have been thus interesting to refer to these publications and to analyze PP7VP1u internalization in brain, heart or mesenchymal cells. 

As suggested, we have added a sentence and the corresponding citations in the introduction.

L46-48: The internalization of B19V by antibody-dependent enhancement [13,14] may explain the presence of viral components in nonerythroid tissue [15], and the possible association of B19V infection with cardiovascular disease [16,17].

Specific comments:

1) Protocols are not deeply enough addressed to properly understand the scope and the significance of the data presented: the calculation of the uptake percentage in figures 2, 3, 4 and 6 is blur, and the means of 100% uptake is not clearly explained. How many copies are internalized per cell? Also in material and methods, line 191, the quantity of bioconjugates and B19V is not relevant as there is not units mentioned.

In this study, we use different constructs, native virions and different types of cells. To quantify the influence of competitors or the cell type, we performed serial dilutions of the sample with maximal uptake (100%), which is the sample without competitors or the most susceptible cell line. Similarly, for uptake kinetics, we compare the different time points with serial dilutions of the sample with maximal uptake (100%).

Although we do not express the results in copy numbers, the quantification of bioconjugates and virions we apply in the competition and kinetic assays is required.

For the sake of clarity, we have changed the legend in the Y-axis of figure 2A and B to better reflect the nature of the experiments and added a sentence in Material and Methods:

L206-207: Results were normalized by the quantification of the β-actin gene and expressed as a percentage of maximal uptake.

2) In figure 2, recombinant VP1u is used to compete with PP7-VP1u internalization. It would have been interesting to apply increasing range of recombinant VP1u to properly quantify the affinity of PP7-VP1u to the VP1u receptor. 

We previously reported (Leisi et al., 2016, Viruses, 8) that dimerization increases the affinity for the VP1uR. Accordingly, the affinity cannot be compared in this experiment. VP1u is a dimer (two VP1u subunits dimerized with the anti-FLAG antibody), whereas PP7-VP1u is a 25 nm particle with VP1u monomers.

For competition experiments, VP1u is added 40 minutes before PP7-VP1u. Is it known if recombinant VP1u is internalized or did it binds to its receptor in an irreversible manner? This question should be discussed by the authors. A similar question is raised by figure 2C with 100% uptake at 15 minutes. 40 minutes after a first application of PP7-VP1U, are cells able to internalize a new batch of PP7-VP1U? 

No internalization will occur, because preincubation with VP1u or PP7-VP1u was performed at 4°C. This was mentioned in Materials and Methods “Cells were resuspended in 200 µL PBS and incubated with 500 ng of recombinant VP1u or a 10x molar excess of PP7-VP1u for 40 min at 4°C prior to the incubation with PP7-VP1u or B19V at 37°C for 30 min”. However, for the sake of clarity, we mention it also in the corresponding figure legend (Fig. 2).

3) Figure 6 showed the use of simian PP7-VP1u on human erythroid cells. It would have been interesting to show the complementary experiment using human PP7-VP1u on simian, rhesus and pig-tailed macaque cells. 

We agree with the reviewer that similar experiments with permissive erythroid cells from simian, rhesus, and pig-tailed would be interesting. However, erythroid progenitor cells within Epo-dependent differentiation stages or alternative erythroid cell lines obtained from these animals are not available.

Minor comments:

  • Lines 19 & 22: what is the mean of N-VP1u?

Mentioned in the abstract

L19: “N-terminal of the VP1u (N-VP1u)”

  • Line 26: Parvovirus B19 and B19V are two redundant keywords

Human parvovirus B19 is also referred as B19V. Although having the same meaning, they are listed to ensure the finding of this study with any of the two keywords.

  • Line 43: B19V entry route is not only respiratory. Please clarify.

The sentence was changed as follows:

L43: “Following the main entry through the respiratory route…”

  • Line 45: infected EPC leads to erythroid disorders

Sentence change accordingly

  • Line 109: precise the UT7/Epo cell line used in the study

We mention the source of UT7/Epo and KU812Ep6 in “Acknowledgments”

  • paragraph 2.1: no mention of NB324K cells used in the paper

NB324K has been included in section 2.1

  • Line 124: Please clarify HSA

Human serum albumin is now mentioned in the text

  • Line 138: correct trough by through

Correction done

  • Line 192: precise trypsination condition as it is a key point of internalization analysis

The conditions have been included

  • Line 204-210: Immunofluorescence protocol should be precisely described. Please clarify fixation method as cold acetone/methanol. What are the permeabilisation steps to allow detection of internalized VP1 or B19V? The blocking steps? 

Methanol/acetone fixation does not require a permeabilization step as these organic solvents can permeabilize cells

  • Line 197 and all along the paper, PCR analysis (RT-PCR for PP7) should be corrected as qPCR (RT-qPCR)

qPCR and RT-qPCR are now always used, except when we mention quantification or quantitative.

  • Lines 201-202: position of the targeted region of B19V primers? 

The position of the primes have been included

  • Figure 2D, 3B, 3D, 4B, 4E, 6B: scale bars are missing. 

Scale bars are not informative since whole cells are shown. Nuclei stained with DAPI is used as reference. The objective used in the LSM is now mentioned in materials and methods.

L215-216: “Internalized viruses were visualized by confocal microscopy (LSM 880, Zeiss) using a 63x oil-immersion objective”

  • Figure 4 D: statistical analysis of 2 independent experiments is not relevant.

Figure 4D shows the results of three independent experiments

  • Line 338: E instead of D

Correction done

Round 2

Reviewer 2 Report

accepted in its revised form